# Web Publication of Schmitt's Map of Southern Germany (1797)—The Projection of the Map Based on Archival Documents and Geospatial Analysis

**Gábor Timár** [1,*] and **Eszter Kiss** [2]

1 Department of Geophysics and Space Science, Institute of Geography and Earth Science, ELTE Eötvös Loránd University, Pázmány Péter Sétány 1/c, H-1117 Budapest, Hungary
2 Department of National Coordination, Federal Agency for Cartography and Geodesy, Richard-Strauss-Allee 11, 60598 Frankfurt am Main, Germany; eszter.kiss@bkg.bund.de
* Correspondence: timar.gabor@ttk.elte.hu

**Abstract:** This work shows the original projection of a 1:57,600 scale map of southern Germany at the end of the 18th century, produced under the direction of Karl-Heinrich von Schmitt (1743–1805). The sections of the map were scanned and georeferenced as part of the MAPIRE project, and the results are publicly available. In the present work, we use contemporary documents, in particular the books of César-Francois Cassini de Thury and manuscript sketches of the map found in the Military Archive of Vienna, to show that the overall projection of the map is identical to that used in Cassini's survey of France (first half of the 18th century). In the archive, we managed to find the overview sheet on which—in addition to the Paris Cassini coordinate system—the section grid of the Schmitt map was also constructed. This sketch served as the basis for the compilation and copying work, wherein the existing map works and survey sketches were inserted into 197 sections of the Schmitt map. Thus, the map coordinate system can be modeled in GIS systems using the Cassini (or Cassini-Soldner) projection, with the Paris Observatory as the projection origin. The georeferencing accuracy of using the pure Cassini projection is around 1–1.3 km (at the extremes, around 5 km), which is much more inaccurate than the one used in later topographic surveys. It is considered a combined result of the compilation of the different maps, presumably surveyed by graphic triangulation with measuring tables.

**Keywords:** georeference; historical cartography; map projection analysis; Schmitt'sche Karte; Germany

## 1. Historical Background

The first wave of Napoleonic Wars was interrupted by the Peace of Basel (1795), and then concluded by the Treaty of Leoben (April 1797) and the Peace of Campo Formio (October 1797) between France and the Habsburg Empire. The Peace of Basel promised compensation to Prussia on the right bank of the Rhine for losses suffered on the left bank. France gained more territory at the expense of the Habsburgs (in the Austrian Netherlands and some Italian and Adriatic territories). The treaty fixed the French borders with Germany on roughly the same lines as today. In what is now southern Germany, including the present province of Salzburg, there were a series of small and medium-sized states that cooperated with Austria in the German Confederation. This federal relationship formed the basis for the mapping of southern Germany, which has resulted in the map work discussed in this paper. The survey was carried out by Austrian military cartography in 1797/1798. The result was a state-of-the-art map of the southern non-Austrian areas of the German Confederation, which was suitable for planning any military operation. In the Second Coalition War of 1799–1802, the main Franco-Austrian front was still in northern Italy. However, the operations of the Third Coalition War (1805) were already partly fought in southern Germany. Interestingly, the Battle of Dürnstein (Lower Austria),

which resulted in Schmitt's death, was also a direct consequence of the battle of Ulm in the mapped area. The Habsburgs quit this war in 1805. In 1806, Napoleon created the satellite state organization known as the Rhine Alliance, and later, important military operations took place outside the mapped area.

## 2. The Schmitt's Map

The Austrian Emperor Franz II, who came to the throne in 1792, immediately realized that the Franco-Austrian conflict, which at that time was still mainly threatening northern Italy and the German Lowlands, could spread to Bavaria and neighboring areas. He therefore issued an order as early as 1792 for surveying the area. The scientific and technical basis for this was the First Military Survey. However, the field survey covering the southern part of what is now Germany and the province of Salzburg in Austria could not be carried out on a regular basis due to the war and the foreign territory. Colonel (later: Quartermaster General) Heinrich von Schmitt (1744–1805) was commissioned in April 1797, after the Treaty of Leoben, to collate the survey data and to survey the missing areas and put them all together into a unified map product. The specific commission was given to Schmitt by Archduke Charles II, the brother of Franz II, at the siege of Kehl. Archduke Charles was deployed as a military commander against the French in the southern German territories, and after the Treaty of Basel, he was transferred to the Rhine region. As Schmitt was appointed head of the General Staff in 1796, he was given the task of leading the mapping. The survey had to be carried out very quickly, as the left bank of the Rhine was inaccessible to the Austrians six months after the order was issued, after the Peace of Campo Formio [1–3].

The main tools used for the survey were the measuring table and the graphic triangulation [4]. The scale of the hand-drawn and colored map work is 1:57,600 [5]. In addition to Bavaria, the map shows the territories of Württemberg, Baden, Hesse-Nassau, Palatinate, and Salzburg (Figure 1) [6]. The 197 map sections showing the entire area were completed by 1798, in the case of the southern part under the direction of Bavarian Colonel Adrian von Riedl (1746–1809). The manuscript map is available in the Military Archives in Vienna [2].

The scale of the sections is 1:57,600, which is half the scale of the Habsburg survey and one and a half times the scale of the Cassini survey of France. The terrain depicted on the map sections measures 19,200 × 12,800 Viennese fathoms, or 36,412 × 24,275 km [1]. Perhaps to aid in the drawing of the content, the sections are marked with auxiliary lines that divide the map content into 6 × 4 squares, with a field size of 3200 × 3200 Viennese fathoms. The map content of the sections measures 63 × 42 cm.

Today's countries and provinces covered by the map are as follows:

- Germany: Baden-Württemberg (full coverage), Bayern (almost full coverage), Hesse, Rhineland-Palatinate (cc. half of their territories), and some parts of Saar and Thüringen;
- Austria: Salzburg and Vorarlberg (full coverage), and some parts of Tirol;
- Lichtenstein: full coverage;
- Some parts of France and Switzerland.

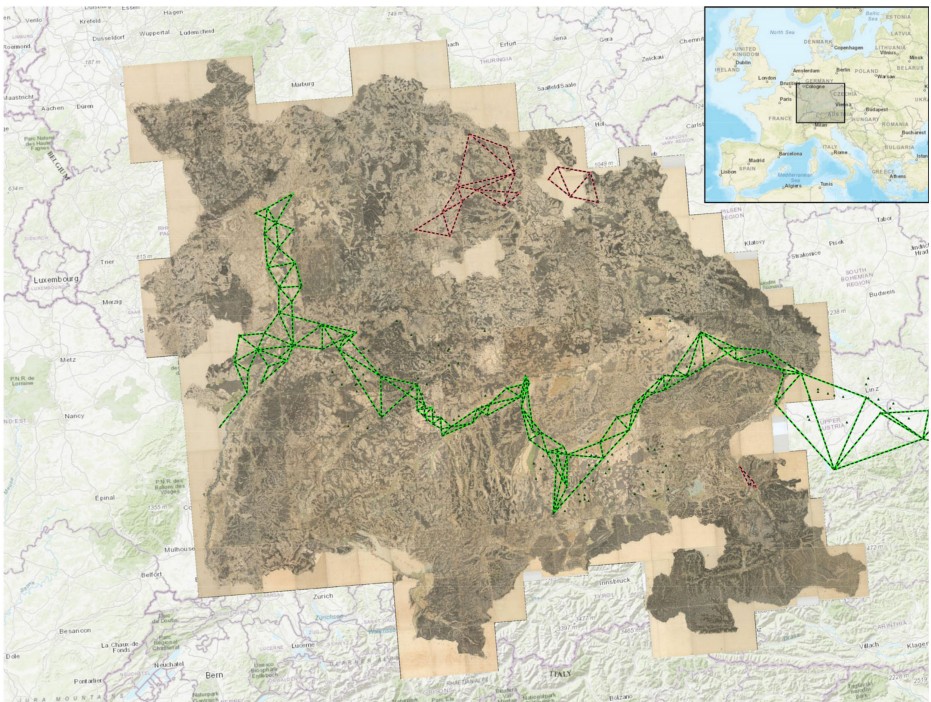

**Figure 1.** Extents of Schmitt's map as a georeferenced overlay on a modern map database (ESRI topographic). The green lines indicate the Cassini triangle chains of the early 1760s from France to Vienna and Frankfurt (this refers to Frankfurt am Main). Purple lines indicate the local surveys made by Cassini, not connected to the main triangulation chains.

## 3. Previous and Contemporary Survey Maps

Similar cartographic works of the period were all modeled on the technology of the map of France by César-François Cassini de Thury (1714–1784), surveyed in the first half of the 18th century and published from 1744 [7,8]. The geodetic basis of the Cassini map was a triangulation network of more than 400 points, spaced at a distance of about 10–40 km [9]. The angles in the network were measured to the nearest 10 arc seconds. The fundamental point of the network was the Paris Observatory. Using this data and a baseline measurement, the map plane coordinates of the points were determined.

The first step of this calculation was to reduce the angles between the other points of each triangulation—the sum of which, because of the curvature of the Earth, was usually, as Schnell & Hortensio had already recognized [10], slightly more than 360 degrees—proportionally so that their sum gave the 360 degrees in the plane. From there, using one of the measured triangle sides, the so-called baseline, the sides of the triangles were calculated by the cosine theorem. The orientation of the planar coordinate system was set by astronomical measurement of the azimuth of a triangle side pointing from one fundamental point to another. The algorithm described in this paragraph is the one later formulated into equations by Johann Georg Soldner (1776–1833) of Bavaria; it is called the Cassini, or—in ellipsoidal case—(Cassini-) Soldner projection [11].

The direct cartographic antecedent of the Schmitt map was the First Military Survey of the Habsburg Empire [12]. It was entirely based on the French technology. The transfer of technology took place in Vienna during the Venus transit of June 1761, when César-François Cassini and Joseph Liesganig (1719–1799) jointly observed and triangulated the area around Vienna [13,14]. The fundamental point varied from province to province: for example, it was Vienna in the case of Lower Austria [15]. In the case of the provinces mapped at the beginning of the survey's two-decade history, the grid of sections does not point in the direction of geographic north at the fundamental points; here, a rotation in the direction of magnetic north at the time is assumed [16].

It is also true for the French and Austrian maps that, compared to the previous maps which still used only astronomical base points, their overall accuracy is much better. It means that they can be georeferenced over most of the mapped area with an accuracy of a few hundred meters (1.5–2 km in extreme cases), assuming the projection starting point is known. This is an improvement not only in accuracy but also in homogeneity of accuracy compared to the previous technology [15]. These errors of a few hundred meters or 1–2 km are not caused by the inaccuracies of the triangulation: an angular measurement accuracy of 10 arc seconds would only cause an error of 1 m at 20 km. Even the conversion to planar coordinates does not make this much worse. The real reason lies in the method of measurement: the accuracy of the so-called graphic triangulation with the measuring table and its associated telescopic ruler is significantly lower than that of measurements at triangulation points.

Apart from the mentioned French and Austrian maps, such surveys were also used in other parts of Europe in the late 18th century. In Denmark, Thomas Bugge (1740–1815) and Caspar Wessel (1745–1818) [17,18], in Norway Bugge [19,20], in Britain William Roy (1726–1790), and other parts of Germany, Friedrich Christoph Müller (1751–1808) and Georg Christian von Oeder (1728–1791), among others, carried out such surveys [6]. All of them were homogeneous and of good accuracy compared to earlier maps but of poor local accuracy compared to later topographic surveys. The triangulation basis is also known for the early Danish and German maps [6,18].

## 4. Triangulation Works in the Schmitt's Map Area before 1797

After the survey of France, taking advantage of the alliance system of the Seven Years' War period, Cassini continued his survey towards both the Austrian Netherlands and also the German and Austrian territories along the Danube and Rhine [21,22] Starting from Strasbourg, which he surveyed in the French mapping, he continued his triangulation through the Württemberg highlands to the upper Danube. From there, it continued largely along the river and south of it in the Munich area, reaching Vienna by 1761, the time of the Venus crossing mentioned above [13]. The survey technology was the same as in France. The data of the survey triangles, the calculated plane coordinates of the points, and a precise description of the calculation method can be found in Cassini's books. These books are now available free of charge on Google Books. The plane coordinates given are in toïse units [23], interpreted from the Paris Observatory in Cassini projection (Figure 2). They can thus be considered an extension of the French survey, not only in technology but also in the coordinate system.

Cassini also carried out more surveys in German territory until the early 1760s. The Rhine survey chain also started in Strasbourg and extended northwards to Frankfurt and Mainz; the plane coordinates were also given in the Paris system. He also surveyed two larger and one smaller separated, 'island' networks, not linked to the above-mentioned chains. The two larger ones are located along the river Main: the one to the west covers the areas of Würzburg, Bamberg, and Coburg, and the one to the east covers the neighborhood of Bayreuth. The smallest 'island' consists of just a few points and covers the immediate area around Salzburg in modern Austria. Since the latter surveys do not connect at any point with the Danube and Rhine chains connected to the Paris system, the plane coordinates are not given at all [22].

In connection with the triangulation along the Danube, Cassini also started a survey in southern Bavaria, complemented by a baseline survey, on the initiative of Bavaria. The work was continued by Henri de Saint Michèl (?−1793), under whose leadership two 1:86,400 scale maps were completed of Munich and further north, the area between the Isar River and the Danube. He also produced a smaller-scale overview map of Bavaria and Palatinate-Neuburg. An atlas of Bavaria was produced by the aforementioned Colonel von Riedl. As this does not include overview maps, but only depicts the areas along the main roads, it would be better described today as an itinerary. Some kind of survey was also necessary for the production of these maps, but the authors have no detailed information

on this. Colonel von Riedl was also involved in the process of compiling the maps of the Schmitt compilation by inserting his finished maps [1].

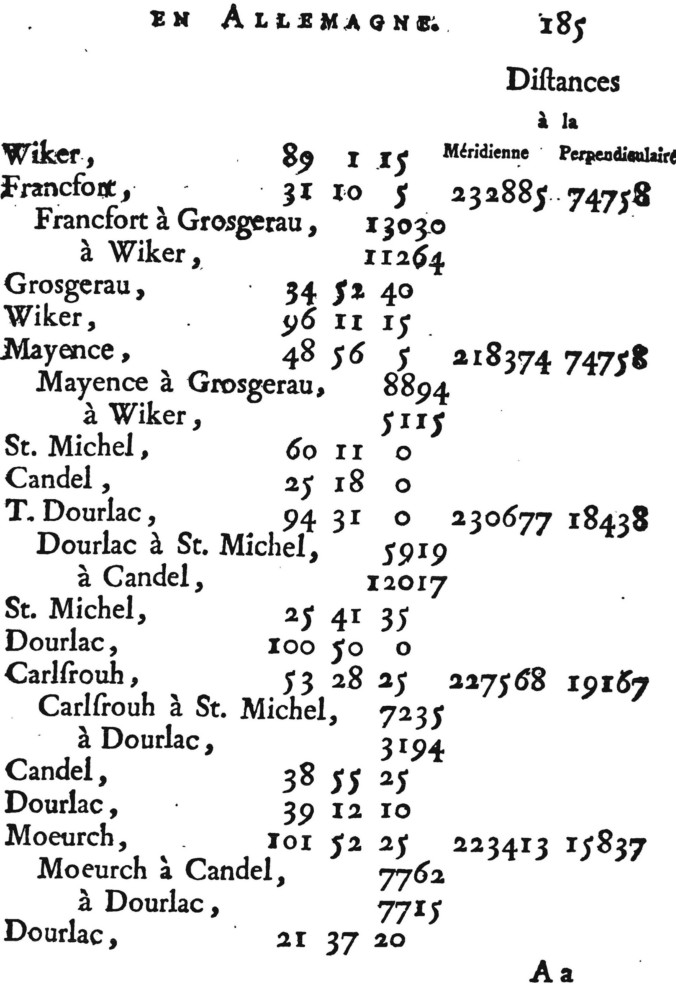

**Figure 2.** Part of a triangulation point list of C-F. Cassini, published in his book [21]. The whole dataset was published in two separate books, available at the time of the survey. The left columns show the observed angles, adjusted to the sum of 360 degrees (see text). The two right columns are the calculated coordinates of the points in the Paris-centered Cassini "projection", in toïse units.

More triangulations were carried out in the western part of the area shown by Schmitt's map. Between 1787 and 1797, Ignatz Ambros von Amman (1753–1840) surveyed the Augsburg area. Connected to this from the west, Johann Gottlieb Friedrich Bohnenberger (1765–1831) surveyed Swabia from 1795 onwards, based on the baseline taken at the Tübingen observatory and its vicinity. Christian Mayer's Palatinate survey was carried out between 1763 and 1772, also in close collaboration with Cassini's work [6]. What the works mentioned in this paragraph have in common is that their triangulation points are also included, to a lesser extent, in Cassini's point list. Detailed data on these surveys are not known to the authors.

The above-mentioned surveys completely omit the north-northeastern part of the Schmitt map. However, the Staržinsky–Sarret survey, which can be regarded as one of the geodetic bases of the Schmitt map, is known for this area [24,25]. This can be seen as a triangulation chain connecting Cassini's Rhine and Danube chains through the 'island-like' surveys along the Main River, roughly along the Frankfurt–Würzburg–Coburg-Nürnberg–Regensburg axis, extended also to the area around Bayreuth (Figure 3) Although neither the surveyed angles nor the calculated plane coordinates are known, detailed sketch maps show which points were included in the survey. The triangulation points are only shown

on the available sketches at the western end of the survey chain, in the Eifel region and roughly between Frankfurt and Würzburg. To the southeast of this, the points included in the actual triangular survey can only be assumed to be located on the elevations, major towers, and Cassini 'island survey' points identified on the topographic maps.

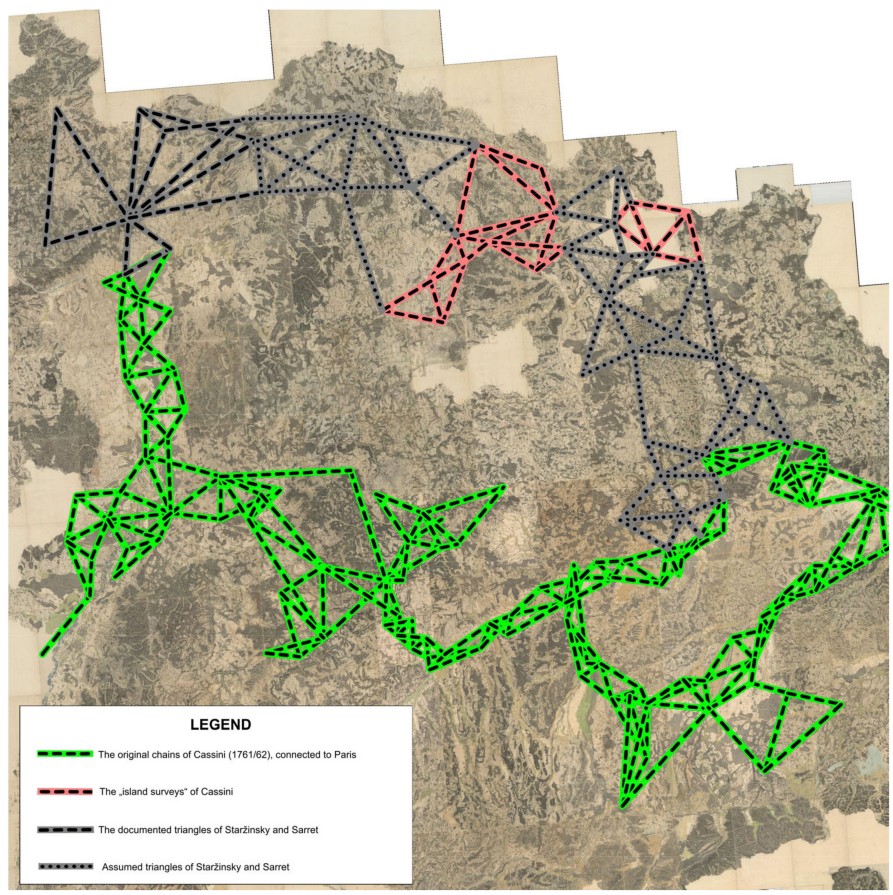

**Figure 3.** The survey of officers Staržinsky and Sarret, connects the Cassini's Rhine and Danube chains, via the separated networks of Cassini along the Main River. The western part of this survey is verified as a triangulation chain (dashed lines; cf. [2]), while the dotted line indicates triangles assumed to be surveyed by triangulation, according to the modern SRTM elevation data.

## 5. The Construction and Projection of Schmitt's Map

The map was compiled by assembling the topographic maps and sketches of the area up to 1797 and by carrying out a rapid survey of the missing areas. According to the literature, the source and accuracy of the maps and sketches used in the compilation also varied widely [1–3]. They also made use of survey material from Swabia and Bavaria and von Riedl's atlases of routes and rivers in Bavaria. Both the existing sketch maps and the rapid survey sketches were probably prepared by means of a measuring table survey, using graphic triangulation, and with very few exceptions had no standard geodetic basis, even less so than is assumed for the first military survey [15].

The most important novelty of the present study is that the authors have found a somewhat incomplete document in the Vienna Military Archives on which the sectional plan of the map to be drawn is mapped (Figure 4). To the best of our knowledge, this document has not been previously described in the literature. This part of the sketch shows the position and numbering of the sections at a scale of 1:57,600 and the location of some landmarks. What is particularly interesting is that it also shows a coordinate system: the Cassini system, starting from the Paris Observatory. This coordinate system is slightly rotated from the sheet boundary grid. Stigloher mentions two overview maps [2], also in the Military Archives in Vienna. One might be similar we mentioned, with the title of

"*Scelett zur Übersicht der grossen durch den Generalstab aufgenommen Karte*" (Skeleton for the overview of the large map recorded by the General Staff), with similar content but not the picture of it. The other one held a handwritten note: 'rotated by 5 (old) degrees relative to the Paris coordinate system'. This document was not found by the authors, but their own verification confirmed the 5-degree rotation between the Paris system and the grid of sheet boundaries. This is important because it can be used to reject any attempt to find the center of a local Cassini projection along a longitude of about 8.34 degrees from Greenwich, as estimated from the sheet grid directions by e.g., Molnár and Timár [26].

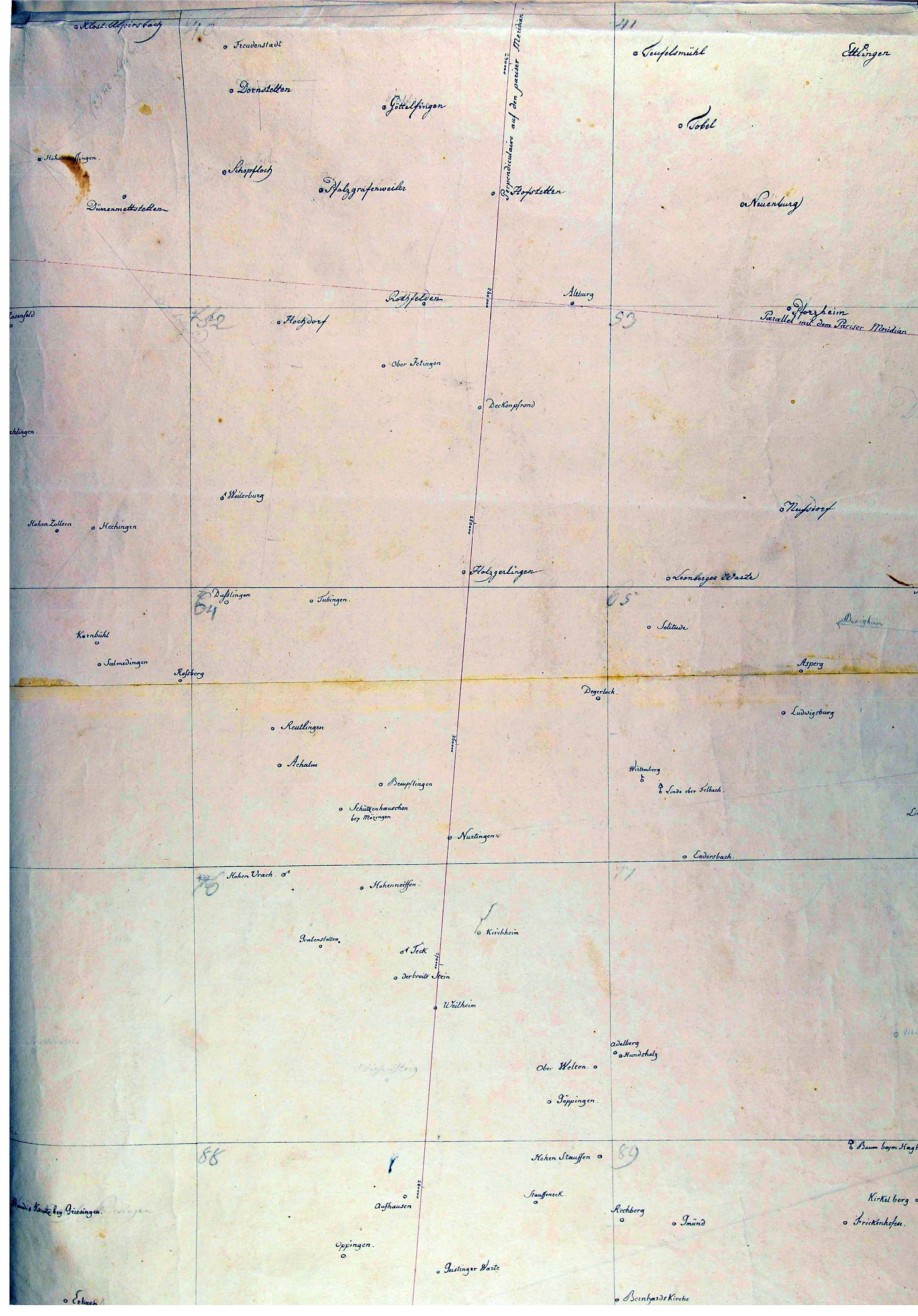

**Figure 4.** The sketch, controlling the collate work performed by Schmitt's staff (military archive in Vienna, drawer BaIVa-72-1). The outlines of the later compiled sheets form a rectangular grid. The Cassini coordinate system (cf. Figure 2), indicated by the lines with texts of "*Perpendiculaire auf den pariser Meridian*" and "*Parallel mit dem Pariser Meridian*", is rotated 5 degrees clockwise from the sheet boundary grid, according to the findings by Stigloher [2]. The base points of Cassini, as well as other local terrain points, are also indicated.

This sketch shows the eastward axis of the Cassini projection taken from the Paris Observatory, and a line parallel to the Paris prime meridian in this projection. On the first axis, the distances from Paris are also indicated by numbers in units of 10,000 toïses, while on the second axis, only small tics in units of 10,000 toïses are shown. In this coordinate system, the positions of the triangulation points given in the Cassini books [21,22] in the same units and system can be easily indicated. The fact that Schmitt's map is roughly in the Cassini projection is a consequence of this very technique. It was through this projection that the overview sketch was made. The compiled maps and sketches, which are in what we now call 'local systems', were manually redrawn into this system. Thus, overall, the error of the Cassini fit was determined more by the accuracy of the local surveys and the accuracy of the compilation to the Cassini frame than by the consistency of the basepoints of Cassini (or the Staržinsky–Sarret point grid).

## 6. Scanning, Georeferencing, and Web Publishing of Schmitt's Map

The original, manuscript version of the map product is available in the map archive of the Military Archives in Vienna (BaIVa-72-1). In the framework of the MAPIRE (now Arcanum Maps) initiative [27,28], the overview page and 197 sections of the map product, plus 13 fold-out sub-sections, were scanned by Arcanum staff in 600 dpi resolution, lossless TIFF format. This version is preserved for digital archive purposes, but for later processing and web publication, slightly compressed JPG files with a resolution of 300 dpi, were used. The final files are in georeferenced JPG2000 format with the coordinate system of Web Mercator (EPSG 3857).

At the time of the georeferencing, carried out in 2017, the information described in the above points was not yet known. The first step was to create a virtual mosaic of 197 complete and 13 partial sections. On this map mosaic, 130 ground control points (GPCs) were selected. These were used for georeferencing using the GSB-based procedure [29,30] of the Global Mapper v20.1 software. This method was quicker and still more accurate than other multi-sheet applications, e.g., [31,32]. As a result, the system of section boundaries in the mosaic is slightly distorted compared to the original regular grid, but the map content is still the best fit for the present-day map. The accuracy of the alignment is, therefore, a few hundred meters and could be further improved by increasing the number of GCPs used. The resulting georeferenced file is the result of this process and is publicly available on the MAPIRE portal (Figure 5). The georeferenced raster map mosaic is a good starting point for vectorization procedures [33,34].

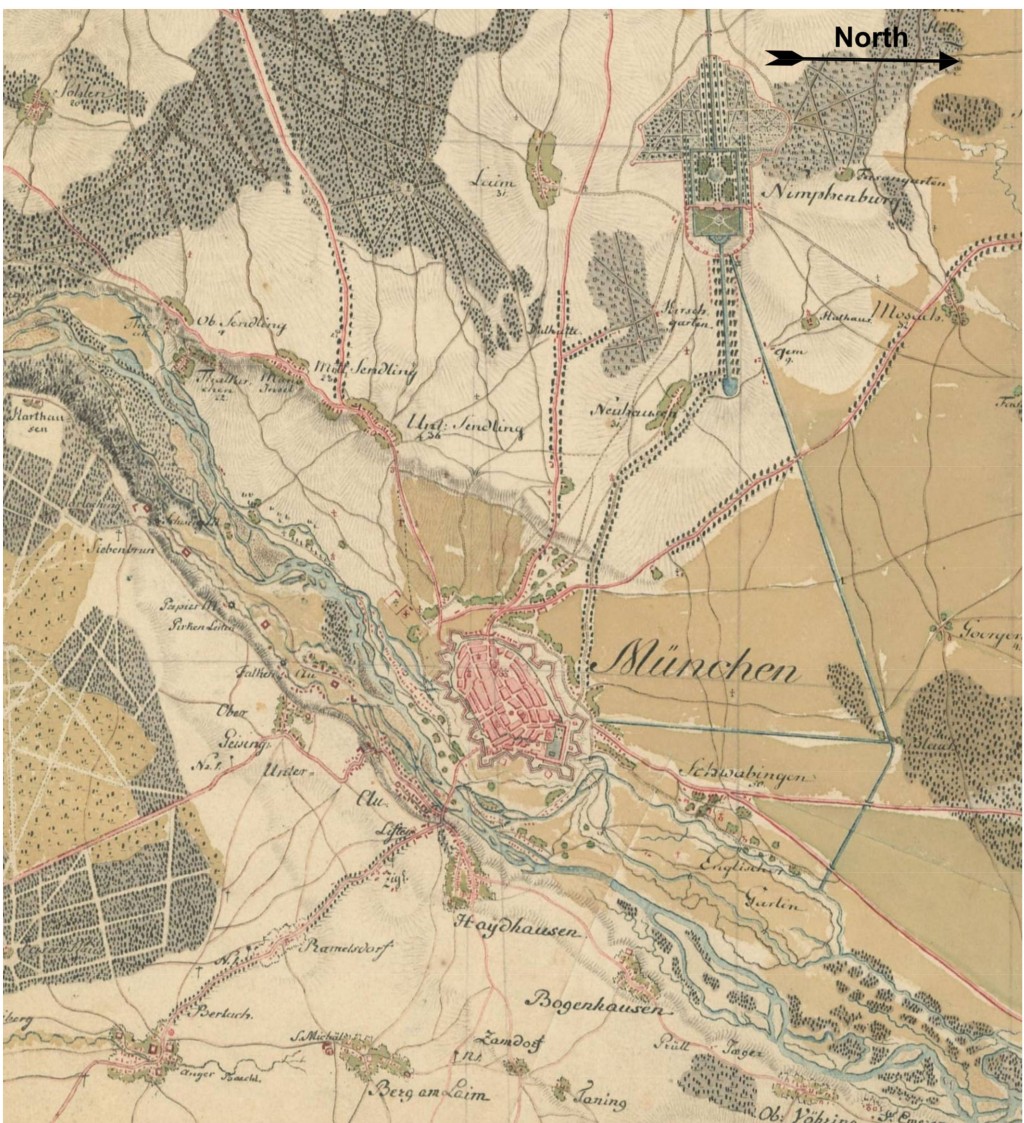

**Figure 5.** Munich and its vicinity (nowadays the downtown of the city) on the Schmitt map. The whole map mosaic is available on the MAPIRE portal [28] in georeferenced form.

## 7. Accuracy of the Georeferencing of Schmitt's Map

The horizontal accuracy of maps to their own coordinate systems has been studied since the 19th century. Initially, line offset vectors were used to assess accuracy (e.g., [35–37], later supplemented by cartographic diagrams drawn as circles or ellipses (e.g., [38,39]). Nowadays, distortion grids are increasingly used [40–43], and choropleth and isorhythmic maps based on deviation data are also encountered [44]. For georeferenced maps, the error analysis is done to today's reference maps (OpenStreetMap, ESRI Topography) and the accuracy of the fit is characterized by the above methods (cf. [45]). In our case, the most useful and also the most spectacular method is to investigate the distortion grid of the sheet boundary network.

If the map mosaic is georeferenced in a Cassini projection with a Paris origin, the output map will still have a rectangular, undistorted grid of sheet boundaries. The georeferenced map, however, will not be accurate when overlaid on a modern database; there are fitting errors of up to kilometers; and we want to quantify and show exactly these. However, the result of the georeferencing based on 130 GCPs used in the MAPIRE processing is that the sheet boundaries are distorted with respect to the original regular grid, but the fit of the map content is optimized according to the given GCP set (Figure 6).

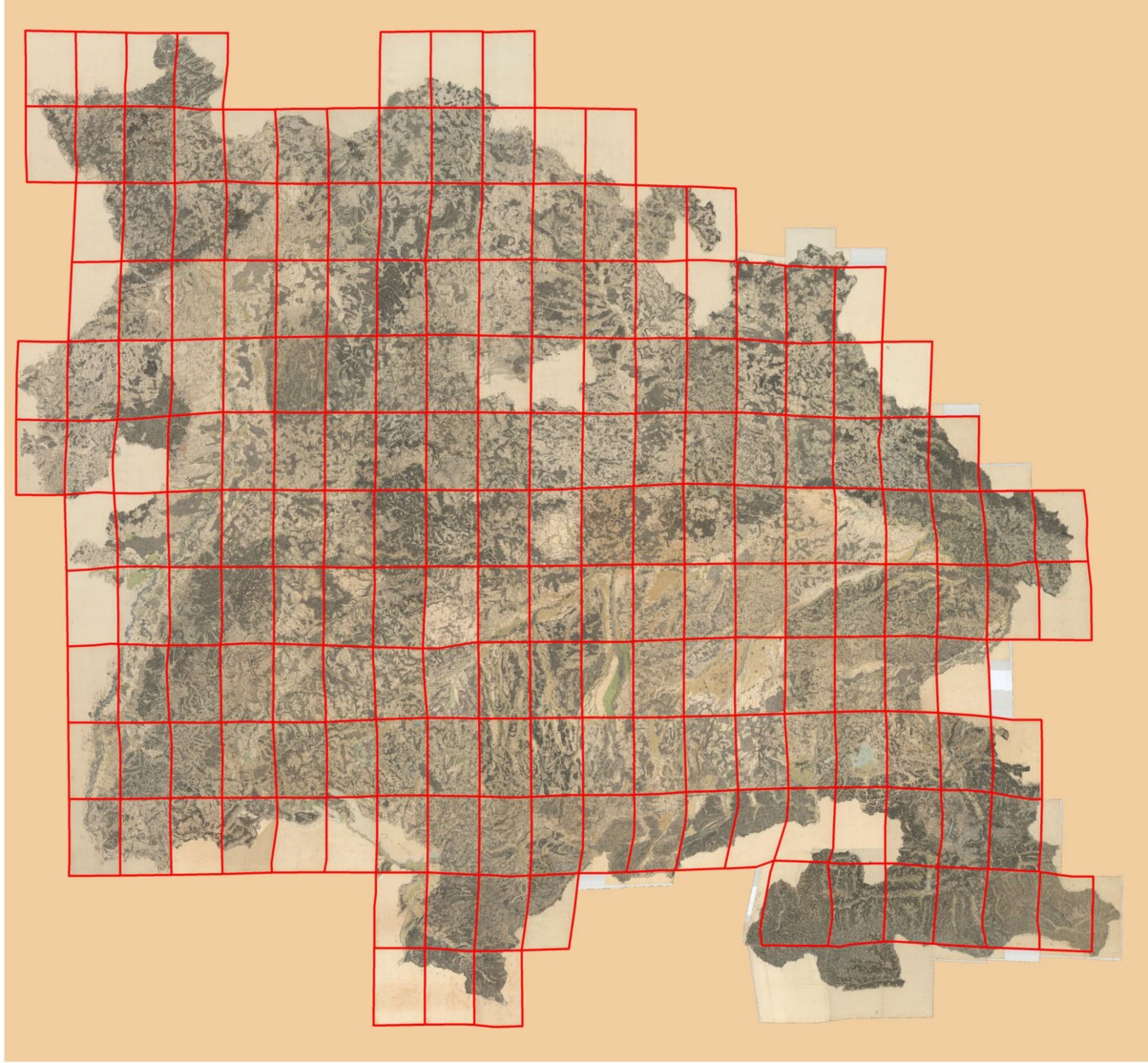

**Figure 6.** The sheet boundary grid, originally rectangular in the Cassini system of the original map mosaic, becomes distorted after the GCP-based georeferencing. This process distorts the sheet grid but provides an optimum fit to the terrain, as shown by modern databases. The magnitude of distortion shows the error of the Cassini projection model of the map coordinate system (cf. also Figure 7).

The difference in the position of the GCPs between the two types of georeferencing: the Cassini linear fit and the local bias based on 130 GCPs can also be quantified. The following error map has been produced by interpolating these deviation data and shows the same numerical values—if not as spectacular—as the sheet grid distortions (Figure 7). Some correlation between the size of the correction vectors and the location of the Cassini triangular chains can be indicated: for example, in the part of the Danube chain consisting of small triangles, the local correction demand is spectacularly reduced. This is not yet a general rule but may be explained by the fact that there were more terrain control points available for redrawing a section than were actually available in the coordinate system of the future mosaic.

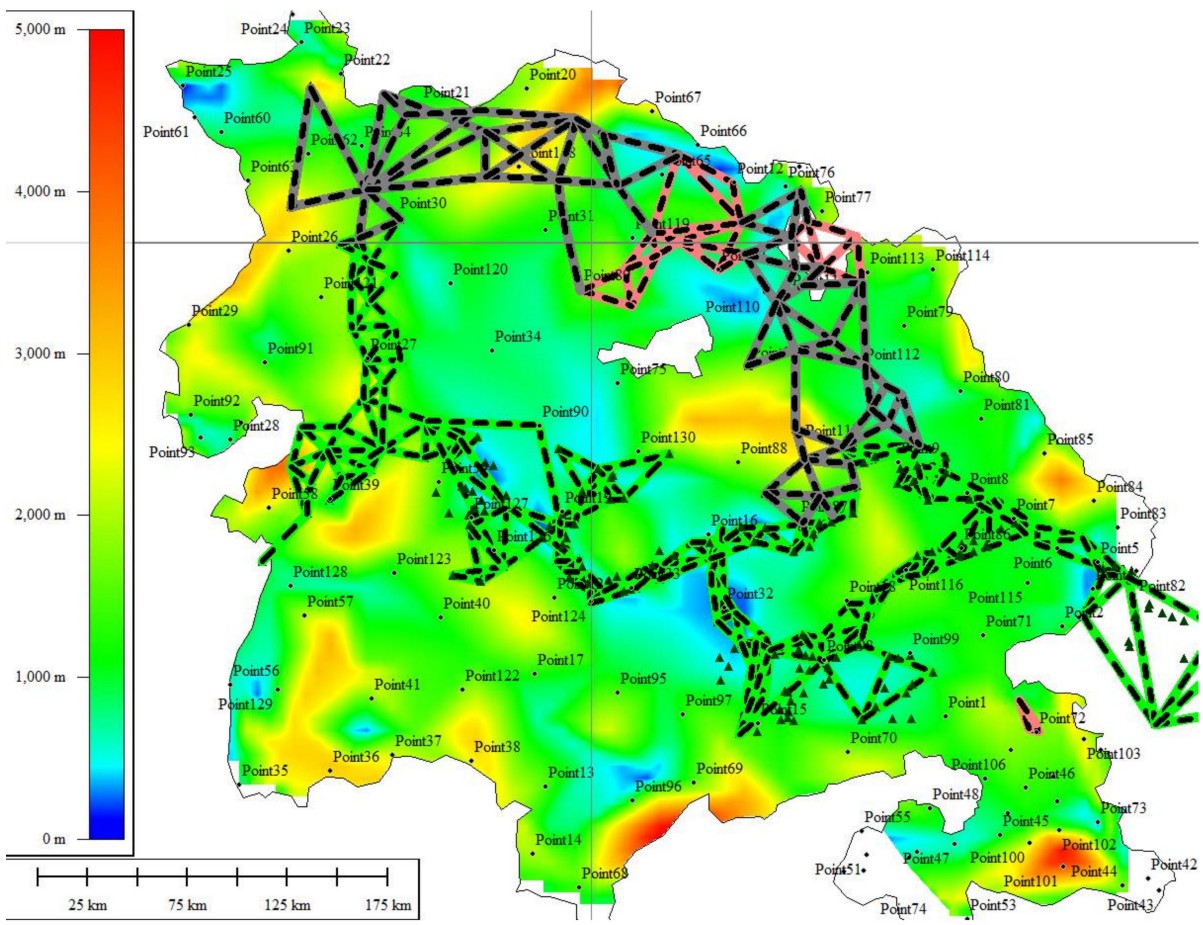

**Figure 7.** The error distribution of the georeferenced Schmitt mosaic: the horizontal distance of the 130 GCPs in the original Cassini projection and in the corrected mosaic. This indicates that even the early geodetic surveys helped to keep the model error relatively low, even throughout such a collating work as Schmitt's map. Note that at a map scale of 1:57,600, a field error of 1000 m equals to 1.7 cm shift on the map. Color codes of triangulation chains are explained in Figure 3.

## 8. Conclusions

1. The projection of the Schmitt map, like the two main triangulation chains, is the Paris-centered Cassini projection, from which the network north is rotated by 5 degrees, thus coinciding the geographical and network norths in West Germany;

2. The projection mentioned in the previous point was achieved in practice by using a controlling, low-scale map sketch in the Cassini projection;

3. The map content of each sheet was compiled from previous maps, sketches, or rapid survey data according to this control sketch. The majority or all of the original map material is assumed to have been prepared in local coordinate systems using a measuring table by graphic triangulation.

4. The Cassini-based control sketch ensured that the map did not show the major regional distortions that characterized maps of similar size until the early 17th century. If the section mosaic is georeferenced in the Cassini projection without further correction, the residual errors average 1–1.5 km, up to 5 km at extremities, partly due to internal survey distortions in the maps on which the compilation is based, but more largely due to hastily carried out redrawing.

5. The nature of the accuracy of Schmitt's map is very similar to that of contemporary cartography: no significant regional distortion errors, but local survey errors are large by today's standards. And not only by today's standards. The local errors are about twice as large as the average error of 5–600 m of the Cassini map of France or the

Hungarian zone (the largest one) of the first Habsburg military survey, which is similar in area to the Schmitt map. This is the difference between the accuracy of regular triangulation-based surveys—but still without subordinate geodetic networks—and map works based on compilations of maps, sketch maps, and survey sketches from different sources.

**Author Contributions:** Conceptualization, Gábor Timár and Eszter Kiss; methodology, Gábor Timár and Eszter Kiss; validation, Gábor Timár and Eszter Kiss; formal analysis, Gábor Timár; investigation, Gábor Timár and Eszter Kiss; resources, Eszter Kiss data curation, Gábor Timár; writing—original draft preparation, Gábor Timár and Eszter Kiss; writing—review and editing, Gábor Timár; visualization, Gábor Timár; supervision, Gábor Timár and Eszter Kiss; project administration, Eszter Kiss. All authors have read and agreed to the published version of the manuscript.

**Funding:** This research received no external funding.

**Data Availability Statement:** The geo-referenced version of Schmitt's map is publicly available on the portal of the MAPIRE project: http://www.mapire.eu (accessed on 1 May 2024).

**Acknowledgments:** The authors are grateful to the partners in the MAPIRE project; the Hungarian SME Arcanum Ltd. and its leaders, Sándor and Előd Biszak, as well as the primary data providers in this project: the war archive of the Österrechische Staatsarchiv and the 2022 Hungarian delegate Ferenc Lenkefi, to this institute.

**Conflicts of Interest:** The authors declare no conflicts of interest.

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
