# Peer review of "Web Publication of Schmitt’s Map of Southern Germany (1797)—The Projection of the Map Based on Archival Documents and Geospatial Analysis"

_ijgi, doi:10.3390/ijgi13060207_

Round 1
Reviewer 1 Report
Comments and Suggestions for Authors
Dear Authors,
First of all I am honoured to review this manuscript. I am a long-time fan of Mapire project and praise the tremendous work done by your team. This paper on Schmitt's map is very interesting and continues covering the state-of-art research, which is valuable and definitely worth publishing.
On the other hand I have listed several minor issues that should be considered before the publication. Please think of them as the constructive criticsim and suggestions to help with the clarity of the presentation of results:
1. Figure 1 presents very good reference of the area covered, however in geographic description only historical states and provinces are used. Please consider adding one paragraph that would describe which moden states and provinces are covered. This is important for global audience that may not be familiar with nuances of 18th/19th century European geopolitics.
2. I suggest refering to Frankfurt as Frankfurt am Main, to distinguish from other eg. Frankfurt an der Oder, etc.
3. Figure captions are detailed and moslty overcomplicated. Please consider adding map legend (fig. 3), north arrow (fig. 5) to subside lenghty captions.
4. I cannot fully agree with the first sentence in chapter 7 line 275-276. I believe there is a 1.5 century-old discussion on ideas how to present map accuracy and geometric distortions, starting from Wolf (1879), Walser (1896) and Merczyng (1913), through Pietkiewicz (1960, 1980), to Jenny (2007), Nicolai (2018), Strzelecki (2016), Kuźma (2021). I suggest extending paragraph that would refer to at least several modern publications and describe most common examples form MapAnalys and GIS software.
5. In fig. 7 several suggestions to redesign.
A) Horizontal scale should be relabelled from 0 to 150 km, values used are not intuitive and hard to follow.
B) Vertical scale should have dual labels - one for real values (meters in field) and second for relative values (milimeters on the map, eg. 5000 m -> 8.7 mm). The value of 0.2 mm on the map should be marked as bare-eye perception threshold. This suggestions were used by Bozzano et al. (2024) with satisfactory results.
C) This is optional - spectral Colour scheme used is methodicly improper. There is one variable therefore sequential scheme should be used. If authors insist on using diverging scheme, please consider calculating and marking mean/median error and using two-colour diverging RdBu. ColorBrewer 2.0 is helpful providing hex numbers for it https://colorbrewer2.org/#type=diverging&scheme=RdBu&n=9
6. Some minor bugs highlighted in PDF file (attached).
Looking forward to see the paper published.

Author Response
Thanks for your detailed and focused review. Here I summarize, how we applied your suggestions during this review round:
- as you suggested, we’ve added a new list at the end of the Chapter 2 about the modern-day countries and regions covered by the Schmitt’s map
- at the first mention of Frankfurt, we clarified that it is Frankfurt am Main
- Figures 3 & 5 were updated according to the remarks and their captions were simplified accordingly
- The main point of the review was the criticism about the lines 275-276. We accepted this suggestion and these lines were replace by a new paragraph accordingly. We also added the suggested references, in fact, the paragraph was compiled around them
- Concerning Fig 7: suggestion (A): completed; suggestion (B): We simply could not apply this suggestion (double scale) in an easy way. We made an addition to figure caption accordingly and also cited the suggested Bozzano et al. paper; optional suggestion (C): in the GIS environment we use, this LUT is not found and cannot be loaded into. Apart from the three above suggestions we also remover the erroneous Vorarlberg part of the map.
- We corrected the minor bugs highlighted in the PDF version
Reviewer 2 Report
Comments and Suggestions for Authors
It is a fascinating article.
The introduction gives a very nice description of the history of mapping in southern Germany in the 18th century.
It would be good to mention other methods of georeferencing multi-sheet map works and compare them with GSB used in MAPIRE.
In Results it would be nice to mention the purpose of georeferencing old maps, the creation of their vector models and their use in GIS analyses.
Author Response
Thanks for your kind review. You asked to make some mentions about the alternative multi-sheet georeferencing works (different to the ones applied in MAPIRE) and also a short outlook to the possibility to vector model building based on georeferenced maps. We made both of these suggestions, both in Chapter 6.
Reviewer 3 Report
Comments and Suggestions for Authors
The paper: "Web publication of Schmitt's map of Southern Germany (1797) 2 - The projection of the map based on archival documents and 3 geospatial analysis", introduces the research carried out by the working group very well.
The methodologies are correctly set out and the results amply explained.
Here are just a few suggestions to improve the paper:
1) introduce on page 4 the toise measure but some readers may not know what it is, perhaps it would be better to define it or introduce a bibliographical reference that does.
2) I could not understand (but it is probably a limitation of mine) if independent points for verification (CPs) were used. If they were used it would also be interesting to see a map of the distribution of such residuals. Consider this only as a suggestion not central to such work.
3) I would expand the bibliography perhaps by citing articles that use GCPs and CPS on historical maps.
Best regards
Author Response
Thank you for the kind review and suggestions made to our paper. You’ve made 3 specific remarks. We applied them as follows
- We added a reference about the old French toise unit at its first appearance in the text
- We answer that no GCPs were used just for verification. In case of the original GSB procedure (applied in MAPIRE) it would be misleading, as in GSB / NTv2 technology all used GCPs act as control of local setting. In case of the linear/Helmert fit of the whole map product to Cassini projection it could have been tried but the whole fit was quite robust and the local horizontal errors (the ones the paper is mostly about) are not really influenced on the selection of the 8-10 well-selected GCPs.
- We added some of these works – as Review #2 asked also (Chapter 6).